

# Revisiting internal gravity waves analysis using GPS RO density profiles: comparison with temperature profiles and application for wave field stability study

Petr Pisoft[1], Petr Sacha[1], Jiri Miksovsky[1], Peter Huszar[1], Barbara Scherllin-Pirscher[2], and Ulrich Foelsche[3]

[1]Department of Atmospheric Physics, Faculty of Mathematics and Physics, Charles University, V Holesovickach 2, 180 00 Prague 8, Czech Republic
[2]Zentralanstalt für Meteorologie und Geodynamik (ZAMG), Vienna, Austria
[3]Institute for Geophysics, Astrophysics, and Meteorology/Inst. of Physics (IGAM/IP) and Wegener Center for Climate and Global Change (WEGC), University of Graz, Graz, Austria

*Correspondence to:* Petr Pisoft (petr.pisoft@mff.cuni.cz)

**Abstract.** We revise selected findings regarding the utilization of Global Positioning System radio occultation (GPS RO) density profiles for the analysis of internal gravity waves (IGW), introduced by Sacha et al. (2014). Using various GPS RO datasets, we show that the previously detected differences in the IGW spectra between dry temperature and density profiles are found only in the one specific data version that was used for the original study mentioned above. The differences between

temperature and density perturbations do not have any physical origin and there is no information loss of IGW activity due to the GPS RO retrieval. We investigate the previously discussed question of the temperature perturbations character when utilizing GPS RO dry temperature profiles, derived by integration of the hydrostatic balance. Using radiosonde profiles as proxy for GPS RO, we provide strong evidence that the differences in IGW perturbations between the real and retrieved temperature profiles (which are based on the assumption of hydrostatic balance) include a significant nonhydrostatic component that is

present sporadically and might be either positive or negative. The detected differences in related spectra of IGW temperature perturbations are found to be mostly about $\pm 10\,\%$.

     The paper also presents a detailed study on the utilization of GPS RO density profiles for the characterization of the wave field stability. We have analyzed selected stability parameters derived from the density profiles together with a study of the vertical rotation of the wind direction. Regarding the Northern Hemisphere the results point to the western border of the

Aleutian High where potential IGW breaking is detected. These findings are also supported by an analysis of temperature and wind velocity profiles. Our results confirm advantages of the utilization of the density profiles for IGW analysis.

## 1 Introduction

Internal gravity waves (IGWs) play an essential role in atmospheric dynamics: they couple different atmospheric layers by their angular momentum transport (Egger et al., 2007), they impact large-scale circulation by interacting with the background

flow (Fritts and Alexander, 2003), and, on smaller scales, their breaking leads to turbulent mixing of air (Fröhlich et al., 2007).



Recent papers by Boos and Shaw (2013) or Sacha et al. (2016) found significant effects of the spatial distribution of the IGW forcing. These studies showed that in comparison with a zonally averaged distribution that has been considered by almost all related studies in the past, the longitudinal variability in the IGW forcing leads to distinctly different results like the generation of planetary waves or enhanced Brewer-Dobson circulation.

Although the IGWs are traditionally regarded as small-scale processes, they are often distributed in large-scale hotspots (e.g., Sacha et al., 2015), and are thus able to influence large-scale circulation. All these findings stress the necessity of observational studies and comprehensive knowledge of IGWs and their various characteristics.

IGWs in the atmosphere are studied by a variety of observational techniques based on remote sensing measurements. Among them, the radio occultation (RO) using signals from the Global Positioning System (GPS) has proven to be a powerful tool

that globally provides detailed information about the vertical structure of the atmosphere (e.g., Anthes, 2011; Scherllin-Pirscher et al., 2017). The significance of GPS RO for atmospheric sciences will most likely increase even more in the future, considering growing numbers of planned satellites and related mission projects (e.g., Cook et al., 2016).

GPS RO provides a perfect vertical but worse horizontal resolution of the measured profiles, and the technique is sensitive mostly to IGWs with small ratios of vertical to horizontal wavelengths (Wu et al., 2006). In the uppermost atmosphere, the ex-

ploitation of RO profiles is limited by ionospheric errors and from below, there is a potential problem of artificial enhancement of the wave activity in the tropopause region, when temperature profiles are considered (Foelsche et al., 2008; Schmidt et al., 2008). Therefore, we focus IGW analysis based on GPS RO profiles on the levels above the tropopause up to 35–40 km.

The GPS RO information relevant for IGW studies is retrieved in a series of steps (e.g., Kursinski et al., 1997, 2000; Steiner et al., 1999, 2001; Hajj et al., 2002).The bending angles are retrieved from profiles of the excess phase of Global

Navigation Satellite System (GNSS) signals passing though the atmosphere. Assuming spherical symmetry, the bending angle profiles can be converted into the refractivity profiles. Subsequently, the dry density is derived from refractivity and using the hydrostatic balance and the state equation for dry air, (dry) temperature profiles are calculated (Steiner et al., 1999; Hajj et al., 2002). Although IGWs have been studied almost exclusively using temperature profiles, density profiles can be used too. An original approach to determine key IGWs parameters using vertical profiles of temperature, density, or buoyancy frequency

was suggested in a series of papers by Gubenko et al. (2008), Gubenko et al. (2011) and Gubenko et al. (2012).

Sacha et al. (2014) introduced a novel method for the utilization of GPS RO density profiles. The paper also discussed differences of IGW characteristics obtained either using RO density or RO temperature profiles. The authors found that there is an information loss due to the hydrostatic filtering when the temperature profiles are used. The comparison of perturbations in density and temperature brings also a question of the nature of the temperature profiles derived under the assumption of

hydrostatic balance. Especially, what is the difference between the GPS RO temperature and real temperature perturbations (with nonhydrostatic modes included) in the presence of a broad IGW spectrum? In this paper, we revisit the findings of Sacha et al. (2014), and investigate related questions using different GPS RO datasets as well as radiosonde profiles as proxy for RO.

The utilization of atmospheric density profiles has important advantages because the retrieval of density is not based on additional assumptions that are used in the retrieval of temperature profiles. Density is the first quantity of state gained in the



retrieval process and the hydrostatic balance does not need to be presumed. Moreover, the background density vertical profile is theoretically inferable by means of statistical physics and can be analytically derived.

In comparison with temperature profiles, density profiles can also provide more information about potential IGW breaking by calculation of specific stability parameters (Sacha et al., 2015). This is highly important for observational studies of the dynamical mechanisms that may support theories regarding IGW instability and breaking. The internal gravity wave spectrum observed in the stratosphere is not only shaped by different sources, it also reflects background conditions prevailing in lower layers. This is illustrated by the propagation of orographic gravity waves that are critically filtered when the background wind speed is zero (this condition is fulfilled if the directional shear exceeds 180°, (e.g., Khaykin, 2016)). Due to this filtering effect, orographic IGW modes cannot contribute to observed IGW activity above regions with zero wind speed. On the other hand, regions of small wind rotation at lower levels are often precursors of enhanced IGW activity higher up (Alexander et al., 2009). The theoretical background (Sutherland, 2010) suggests that the IGW should dissipate below the critical level of 180° directional shear rotation. Corresponding research and related discussion are still ongoing. However, analyzing the stability parameters together with the rotation of the wind direction, this mechanism can be properly tested. In the presented study, we have analyzed the vertical rotation of the wind direction together with the distribution of the gradient Richardson number (Galperin et al., 2007; Sutherland, 2010) and $\sigma^2$ representing maximum growth rate of disturbances arising from the Rayleigh–Taylor convective instability (Sutherland, 2010).

The study is structured as follows. The next section introduces an update of the Sacha et al. (2014) study with focus on the comparison of IGW spectra derived from density and temperature profiles, respectively. The results are compared using various GPS RO datasets, showing that some of the previously suggested implications are valid only for a specific dataset. The following part elaborates the question of the nature of the GPS RO dry temperature profiles and their differences from measured temperature. The study utilizes radiosonde profiles with results indicating significant differences between real measured temperature and temperature calculated assuming hydrostatic balance. Analyses of stability parameters and stratospheric dynamics are introduced in Sect. 4. Specific variables derived from GPS RO data are analyzed to describe regions with potential IGW breaking and its connection to vertical rotation of the wind direction. The resulting distribution of the studied characteristics points mainly to the processes at the western border of the Aleutian High.

## 2 IGW perturbations of GPS RO density and temperature profiles

### 2.1 Data and methodology

Sacha et al. (2014) studied GPS RO profiles produced by the Wegener Center Occultation Processing System version 5.4 – OPSv5.4 (Steiner et al., 2009; Pirscher, 2010). Both density and temperature profiles from this specific dataset were analyzed for the 2006 to 2010 time period. In this study we again use the OPSv5.4 data but additionally include profiles from a more recent retrieval version OPSv5.6 ((Schwärz et al., 2016; Scherllin-Pirscher et al., 2017); Angerer et al., 2017, manuscript in preparation for submission in AMT) and also profiles from the COSMIC Data Analysis and Archive Center – CDAAC (Rocken



et al., 2000) and fom the Radio Occultation Meteorology Satellite Applications Facility – ROM SAF (Syndergaard, 2016). All the profiles were analyzed for the same time period (2006 to 2010) applying the same methodology.

For a proper comparison of the profiles and the derived IGW spectra we started with the construction of background profiles. To separate the density perturbations, we applied a method for the background separation based on fitting the buoyancy

frequency height profile with consequent analytical derivation of the background density. To get an appropriate temperature background, we assumed hydrostatic balance and evolution of pressure in the form

$$p_0(z) = \hat{p}_0 - g \int_z^{z_0} \rho_0(z) \mathrm{d}z. \tag{1}$$

Using the state equation we gained

$$T(z) = \frac{p_0 - g \int_z^{z_0} \rho_0(z)\mathrm{d}z}{R \rho_0(z)} \tag{2}$$

for the temperature evolution. To fit the background we iterated perturbed values of pressure and gravity acceleration at the lower boundary. Figures 1a and 1b show the final fit of the background density and temperature from an example profile. By subtracting the background from the original profile we gained perturbations that were subsequently normalized (by dividing the result by the background). The resulting density and temperature disturbances for the example profile are illustrated in the Fig. 1c. As a direct consequence of the application of hydrostatic balance in the GPS RO retrieval, there is a phase shift by

180° between the obtained density and temperature perturbations.

To evaluate differences in the amplitude and frequency of IGW vertical modes we have applied frequency analysis (using the Fast Fourier Transform algorithm, FFT) on the perturbations derived from all the density and temperature profiles. The resulting spectra were compared and the differences statistically analyzed.

## 2.2 IGW perturbations spectra

Figs. 2a-b illustrate power spectra of the normalized density and temperature perturbations and their difference for the OPSv5.4 data. The spectra were calculated as an average from individual profiles over the northern mid-latitudes. While vertical wavelengths larger than about 9 km and smaller than about 1 km are higher for temperature, density perturbations have larger vertical wavelengths between 1 km and 9 km with a maximum of about 20 % at about 2–3 km.

Sacha et al. (2014) found similar differences between IGW characteristics from density and temperature and suggested

that they might be caused by nonhydrostatic IGWs. However, there is no physical explanation why GPS RO temperature fluctuations should have larger amplitudes than density fluctuation because the involvement of hydrostatic balance should only cause a phase difference between temperature and density fluctuations (see discussion below). However, higher values of temperature fluctuations for wavelengths smaller than 1 km could also be connected with noise and wavelengths larger than 9 km could be connected with background fit deficiencies. If these assumptions were true the results would point to limits for

the vertical wavelength cut-offs.

Figs. 2c-d show resulting vertical wavelength spectra using OPSv5.6 profiles as input. The spectra were also calculated as an average from individual profiles over the northern mid-latitudes. The figure shows mean spectra calculated from individual pro-





files. The differences between the PSDs of temperature and density perturbations are significantly less pronounced compared to OPSv5.4 and they are only up to 2 %. The region where the density spectrum has higher values is delimited by different vertical wavelengths than in the case of OPSv5.4. It is bounded by vertical wavelengths of about 1–7 km without a distinct maximum. The very small difference between temperature and density spectra is illustrated also by an analysis of the respective CDAAC

and ROM SAF profiles. Small differences and no distinct maxima were detected in those cases too (Figs. 2i-l). The similarity of the density and temperature spectra might be considered as an evidence of the suitability and accuracy of the method for subtracting similar background conditions.

To more closely study the disagreement among results for the spectra of different datasets, we have further analyzed the differences of the power spectra of the normalized perturbations between the OPSv5.4 and OPSv5.6 profiles. The results

illustrated in Fig. 2e-h show that in case of the OPSv5.6 density profiles there is a significant increase for the modes above 10 km of vertical wavelengths, a slight decrease of about 5 % for wavelengths between about 7 km and 1.5 km and finally a significant decrease for high wavenumbers. In case of OPSv5.6 temperature profiles, the comparison with OPSv5.4 points to an increase in the power spectrum values for most of the wavenumbers. This increase has a maximum of about 30 % for vertical wavelengths around 2 km and is followed by significant decrease for wavelengths smaller than about 1 km.

The results point to a distinct change between the OPSv5.4 and OPSv5.6 and comparisons with CDAAC and ROM SAF show that there is mutual agreement with the OPSv5.6 profiles. The reason for the significant difference between the OPS data versions is not clear. There were minor changes in handling of the background information used in the retrieval, e.g., co-located ECMWF profiles and in the background atmosphere itself in processing the OPSv5.4 and OPSv5.6 data. However those changes could not lead to significant differences in the spectral density of the related perturbations. The answer may

possibly lie outside the processing changes at the Wegener Center and could be connected to different UCAR versions of excess phase and orbit data.

Comparison of the density and temperature disturbances for the example profile in the Fig. 1c points to the phase shift between the perturbations. This difference between normalized temperature and density profiles is in principle theoretically inferable from the ideal gas law under the assumption of negligible pressure perturbations (for details please see She et al.,

1991). The resulting relationship shows that the normalized perturbations are in anti-phase (shifted by 180°). There is also a rather theoretical question going beyond the scope of this manuscript on how this assumption gets along with the hydrostatic balance utilization (see the Appendix of Doswell III and Markowski, 2004).

## 3   Nonhydrostatic forcing and GPS RO dry temperature profiles

In the GPS RO retrieval, dry pressure and temperature profiles are derived from density profiles under the assumption of hydro-

static balance . They would therefore correspond to their real atmospheric counterparts only if the atmosphere were in the state of hydrostatic balance. For IGWs this implies that if the hydrostatic balance is assumed, the temperature perturbations would correspond to actual temperature perturbations only if the disturbances are created by hydrostatic waves only. If nonhydrostatic modes are present too, pressure and density are not hydrostatically linked and the induced dry density perturbations are con-





nected with different temperature perturbations from those that are derived using the hydrostatic balance integration. In this case, the corresponding temperature disturbances could only be computed from a separately measured temperature or pressure profile. To analyze this hypothesis and to quantify the effect of deviation from the hydrostatic balance assumption on GPS RO IGW analysis we have studied proxy data from radiosonde measurements. The PSDs resulting from radiosondes should

be compared to those from GPS RO with caution because of the wavelength distortion due to a different viewing geometry (de la Torre et al., 2017). However in the presented analysis either PSDs from the same GPS RO observations events or from radiosondes are compared. Therefore the effect of the wavelength distortion does not influence the comparison.

## 3.1 Data and methodology

We have analyzed data from the GRUAN radiosonde network (Dirksen et al., 2014; Immler et al., 2010). All available profiles

for altitudes between 15 km and 30 km from the stations in Lindenberg (Germany), Ny-Alesund (Norway), and Tateno (Japan) were included in the analysis, for both temperature and pressure. Density was calculated using the equation of state. Having the vertical profiles of density, we have integrated the hydrostatic balance to acquire pressure and temperature profiles similarly to the GPS RO dry temperature retrieval process. The integration was initiated at the top level using the value from corresponding pressure profiles. Variable gravitational acceleration was included in the calculation. Temperature profiles were subsequently

derived from the pressure profiles using the equation of state.

To verify this methodology applied to GRUAN data, we have applied the very same procedure on GPS RO dry density profiles provided by CDAAC and compared our "retrieved" temperature profile to temperature profiles provided by CDAAC. If our and the CDAAC methodologies were the same in all details, the resulting temperature profiles would be identical. However, in comparison with the raw profiles used for the retrieval we have studied already processed data with vertical extent only up to

20 35 km. To compare resulting temperature profiles and to quantify our methodology error, we calculated IGW perturbations for all our temperature profiles and those from CDAAC (using the methodology introduced in the previous section) and applied frequency analysis. Fig. 3 shows typical power spectra of the normalized perturbations for the derived and original temperature CDAAC profiles, and their difference as a mean over all profiles obtained at mid-latitudes in June 2013. The difference between the profiles is about 2 % for most wavenumbers, though with a maximum over 5 % for wavelengths longer than about 8 km.

Based on these results, we have estimated the methodology error to be about 5 % and applied the algorithm of the temperature derivation to the radiosonde profiles. It is important to note that methodology error could also result from different gravity fields (Scherllin-Pirscher et al., 2017), different (or additional) smoothing procedures or different internal vertical grids.

## 3.2 Comparison of temperature perturbations

Figures 4a-f illustrate resulting spectra of the normalized temperature perturbations for the derived and original radiosonde

profiles, as well as their differences. Although the nonhydrostatic forcing is always present to some degree, significant forcing is assumed to be sporadic. However the perturbations spectra represent an average and the nonhydrostatic impact would be neglected if all profiles were included. To account for this and considering the estimated methodology error, we have selected only those profiles, where the mean difference between the temperature disturbances was above 5 % for vertical wavelengths



shorter than 8 km. Depending on the analyzed station, 10–30 % of the analyzed radiosonde profiles satisfied this criterion. Still, the resulting differences between the perturbations of the original and derived temperature profiles shown in Fig. 4a-f are not significant, ranging from less than 1 % up to 5 %. This might be connected to the fact that there could be nonhydrostatic effects of opposite signs and those might be canceled out in the profiles mean. To test this case we have also analyzed an actual

distribution of the perturbations differences related to individual profiles that were included in the previous analysis step. This is shown in Fig. 4g-i where the color-scale represents the number of profiles falling into specific intervals of perturbation differences and wave numbers. Regarding vertical wavelengths between about 8 km and 1 km, more than 60 % of the perturbation differences are found between $\pm 10$ %. For wavelengths longer than about 8 km, the applied methodology fails and the differences are mostly more than $-40$ %. For wavelengths shorter than 1 km, the differences between the temperature disturbances

are distributed approximately uniformly between $-10$ % and $+10$ %.

Our results point to several interesting findings. There is a significant difference between the IGW perturbations of the original temperature profiles and those that were derived using integration of the hydrostatic balance. These differences can be attributed to nonhydrostatic forcing. Results presented in the previous Sect. 2 showed that the IGW perturbations in GPS RO dry temperature profiles are the same as for the density profiles (except for the phase shift). In case of hydrostatic equilibrium,

temperature (pressure) and density are hydrostatically linked so there should not be a difference between perturbations of derived and actually measured temperature profiles. However, atmospheric processes are of both hydrostatic and nonhydrostatic nature, meaning that temperature and density are not entirely linked by the hydrostatic balance. Hence, the perturbations of the derived temperature still correspond to the density perturbations but they do not represent the whole perturbation spectrum of the separately measured temperature.

The results indicate that the differences between the real and hydrostatic temperature profiles may vanish when all profiles are included in the average, probably due to the sporadic nature of the events with significant nonhydrostatic forcing. Also the nonhydrostatic effect can be masked by averaging due to variable projections of nonhydrostatic modes onto temperature and density. As there can be various sources of nonhydrostatic processes and they may not even be adiabatic, the nonhydrostatic contribution to the temperature can be both positive and negative and vice versa for the density. The reason for the detection of

the positive and negative perturbation differences might then be linked to a dominance of the nonhydrostatic forcing in density or temperature perturbations. If the contribution to density perturbations is positive, the difference between the IGW spectra in the original and derived temperature would be negative. If it contributed positively to temperature perturbations, the difference would be positive.

## 4 IGWs (in)stability using GPS RO density profiles

### 4.1 Data and methodology

To study the wave field stability and to illustrate advantages of the utilization of GPS RO density profiles, we have analyzed two derived IGW parameters introduced by Sacha et al. (2015), i.e. the gradient Richardson number and maximum growth rate of disturbances arising from Rayleigh–Taylor convective instability.



According to Sutherland (2010), the necessary condition for dynamical instability when disturbances overcome the stabilizing effect of buoyancy by drawing kinetic energy from the mean flow, is expressed by

$$Ri_g = \frac{N_0^2}{s_0^2} < \frac{1}{4},$$ (3)

where $Ri_g$ is the gradient Richardson number, $N_0$ the background stratification frequency, and $s_0$ the background wind shear.

Assuming that the wind shear is negligible, scaling the variance of the vertical gradient of density perturbations and using the polarisation relations for IGWs, this can be further adjusted (Senft and Gardner, 1991) to

$$Ri_g = \frac{g^2}{N^4} \left\langle \left[ \frac{\partial}{\partial z} \left( \frac{\rho'}{\rho_0} \right) \right]^2 \right\rangle.$$ (4)

The parameter $\sigma^2$ represents the maximum growth rate of disturbances arising from Rayleigh–Taylor convective instability that is used to describe overturning instabilities. It is defined by

$$\sigma^2 = \frac{g}{\rho_0} \left( \frac{d\rho_0}{dz} + \frac{\partial \rho'}{\partial z} \right)$$ (5)

and its values are real in case that the wave forcing drives the fluid to overturn.

Similarly to the methodology used in Sacha et al. (2015) we did not analyze these parameters to detect exceedance of an exact threshold. Values of the studied parameters Rig and $\sigma^2$ are generally expected to be below the thresholds for the turbulence and mixing to occur because there is a low probability that the GPS RO profile represents exact the time of IGW breaking and also due to the observational filter effect (Lange and Jacobi, 2003). For these reasons we studied the parameters as indicators of potential instabilities in the sense that in case of lower Rig values and higher value of $\sigma^2$, there is higher probability of IGWs breaking and interaction of the waves with the mean state. We have also calculated the mean potential energy density defined in

$$\overline{E}_p = \frac{1}{2} N^2 \langle \xi^2 \rangle = \frac{1}{2} \left( \frac{g}{N} \right)^2 \left\langle \left( \frac{\rho'}{\rho_0} \right)^2 \right\rangle$$ (6)

(Wilson et al., 1991).

To analyze the influence of background conditions, monthly mean climatological wind fields from sampling error-corrected geopotential height fields were calculated using OPSv5.6 GPS RO data. For our purpose, the data offer unique advantages, including geopotential height being accurately obtained as vertical coordinate jointly with an accurate retrieval of pressure, very high vertical resolution, and global coverage. The wind fields were subsequently derived using approach described in Scherllin-Pirscher et al. (2014, 2017) as monthly means in a grid of $5° \times 5°$. For further analysis we have also calculated monthly mean values of $\sigma^2$, Richardson gradient number, and potential energy using OPSv5.6 GPS RO profiles.

### 4.2 Results

For the winter season (December-January-February, DJF), Figs. 5a-b illustrate the distribution of wind direction rotation from ground up to 7 hPa and the distribution of maximum $\sigma^2$. In the northern hemisphere, the wind rotation exceeds 180° mainly





over the Northern Pacific. Maximum $\sigma^2$ is located over East Asia (EA). The geographical distribution of the maximum $\sigma^2$ values overlaps a small part of the western flank of the critical rotation region. This is in accordance with the findings by Sacha et al. (2015).

To examine the possible connection between wind direction rotation and dissipation of IGW, we focused our analysis on
profiles with maximum $\sigma^2$ values that were detected below the 180° rotation level (provided that the 180° rotation was detected at the selected grid point). With regard to the results of the critical rotation study presented in the Fig. 5b, the analysis comprised profiles mainly in the Southern Hemisphere and over the Northern Pacific. Locations of the grid points with maximum sigma squared values detected below the 180° rotation level are shown in Fig. 5c. Selecting only the maximum $\sigma^2$ values that are located below the critical line for IGWs propagating from the surface, our analysis highlighted grid points mainly in the western
boundary region of the Aleutian High (AH). To analyze the vertical structure and longitudinal variability of this region where maximum $\sigma^2$ values are found below the critical rotation level, the follow-up analysis was focused on the vertical profile in the latitudinal belt of 35° N–45° N (highlighted in the Fig. 5c).

Fig. 6 illustrates the resulting vertical distribution of the wind direction rotation and detected maximum sigma squared values together with the surface topography. The averaged maximum $\sigma^2$ values (black crosses) match the critical level locations. The
maximum $\sigma^2$-values from selected individual profiles (gray crosses) are clustered into three groups. The first group below the 25 km level comes close to instability slightly below AH. This is in very good agreement with the GW instability theories stating that GWs break before reaching the critical level (Fritts and Alexander, 2003) and also with connection to the dynamic instability influence (Sutherland, 2010). The second group just above the 25 km comes close to instability directly at the AH border while the third group (approx. above 30 km) tunnels beyond the critical level. These results can be connected to the fact
that the maximum $\sigma^2$-values using individual profiles may not correspond to the average wind rotation characteristic and may be influenced by different AH locations at the exact time of their occurrence (observation). They can be also connected to GWs originating from spontaneous emission from the jet at the boundary of the AH (de la Torre et al., 2006) or secondary GWs created by breaking at the critical level (Fritts and Alexander, 2003). The related topography (depicted at the bottom of the Fig. 6) suggests orographic origin of the IGWs with the source in the EA coastal region. Note that the distance of the Himalayas
is more than 3000 km in horizontal and around 10 km in vertical. Although GWs have been recently found to be able to travel large horizontal distances (e.g., Kalisch et al., 2014), we assume that in this case the almost perfect vertical collocation with the significant EA orography is a more reasonable explanation. Nevertheless, the effect of the Himalayan ridge on the atmosphere is definitely worth additional research taking into account distributions of zonal wind, temperature and Rig (see plots below).

Vertical profiles in the latitudinal belt of 35° N to 45° N for other studied variables are shown in Fig. 7. The position of the
maximum sigma squared is depicted in the same way as in the Fig. 6. The white line represents the borders of region where the wind direction rotation exceeds 180°.

In Figs. 7a-b we can see that the average zonal wind and temperature cross-section is dominated by the planetary wave 1 (PW1) with a typical westward tilt with height and a sharp transition of phase around 150° E. Rig and $E_p$ fields (Figs. 7c-d) both seem to be strongly connected to subsidence, which can be inferred from the temperature field. The fields of extreme Rig
and $E_p$ values are sloping down with longitude in agreement with the downward penetration of the subsidence with longitude,





peaking around 150° E, where the high temperature linked to the subsidence is reaching below 20 km. East of approx. 150° E we can observe an abrupt change to upwelling (visible in the temperature field; (see also Demirhan Bari et al., 2013; Sacha et al., 2016) and also horizontal wind reversal and consequent $E_p$ and Rig disappearing. The maximum $\sigma^2$ values are concentrated along the line of this change. An interesting fact is that the interface between high $E_p$ and Rig values around 150° E does

5   not show any vertical tilt with height (as is the case of zonal wind and temperature), and penetrates the western flank of the AH. This is again a strong indication of the relationship of those quantities ($E_p$, Rig) with the orography lying directly below (Fig. 6). The abrupt end of the IGW activity eastward of approx. 150° E can be attributed to the transition from the continent to the ocean.

## 5   Conclusions

### 10   5.1   IGW perturbations of GPS RO density and temperature profiles

The presented results (Fig. 2) clearly showed that the original conclusions of Sacha et al. (2014) regarding information loss in case of the use of temperature profiles did not prove to be generally valid. Comparison of different RO datasets from WEGC, CDAAC, and ROM-SAF revealed that significant differences were detected only in case of the OPSv5.4 record. This finding is important for the involved community and in connection with all related studies that utilized the OPSv5.4 data in the past. The

15   analysis also pointed to the limits of the utilized methods for the extraction of perturbations. For the wavenumbers below 0.13 significant differences between the spectra of density and temperature perturbations were detected. These results suggest that the analysis fails for vertical wavelengths longer than about 8 km. As for the smaller wavelengths, high similarity of density and temperature spectra was found. This also demonstrates the accuracy of the applied method for the subtraction of density and temperature backgrounds.

### 20   5.2   Comparison of the influence of retrieved and measured temperature on IGW activity

We have confirmed that there are differences between the perturbation spectra of measured temperature profiles and those that are derived assuming hydrostatic balance (used in the retrieval of GPS RO dry temperature). However, these differences are negligible when averaging over a large number (more than one hundred) of profiles. To illustrate the differences it is necessary to analyze distribution for individual profiles as it is shown in Figs. 4g-i. Then, more than 60 % of the perturbation differences

25   are found to be close to $\pm 10$ % for vertical wavelengths between about 8 km and 1 km.

    Considering physical relations among the studied variables, the results suggest that the differences in the perturbation spectra are stemming from nonhydrostatic forcing. Moreover, there is a strong indication that a significant manifestation of the nonhydrostatic forcing is highly sporadic and projection of nonhydrostatic modes on temperature and density can be very different from case to case.



### 5.3 Wave field stability from GPS RO density profiles

We have presented a detailed analysis of the connection between IGW activity, wave breaking and background conditions (zonal wind, its rotation with height, temperature) utilizing data from GPS RO exclusively. High correspondence between the spatial distribution of the $\sigma^2$ parameter and critical rotation of background wind values (Fig. 6) validates the utilization of sigma squared as a wave breaking proxy. This may be of great importance for the IGW observations community as it supplements efforts for estimation of the gravity wave drag (GWD) from pseudomomentum gradients (Ern et al., 2011, 2014) by indicating areas of likely breaking and also regions where the breaking is not likely (and the observational filter dominates).

Our analysis further underlines the importance of the EA/NP region (Sacha et al., 2015, 2016) as a unique source of GWD in the lower stratosphere, showing an important role of the Aleutian High that acts as a robust region of low-level positioned critical wind rotation. The results also pointed to an interesting behavior of the fields of Ep and Rig (Fig. 7) that copy the PW1 structure at the lower boundary (probably due to Doppler shifted IGWs) but lack the typical westward tilt with height at the upper boundary of our analysis.

Based on the presented results we recommend the utilization of density profiles in IGW analyses.

*Data availability.* CDAAC and ROM-SAF occultation data are available upon registration at the websites of the CDAAC: http://cdaac-www.cosmic.ucar.edu/cdaac/ and ROM-SAF: http://www.romsaf.org/registration.php. For the OPS occultation data please contact processing team members at Wegener Center for Climate and Global Change: https://wegcenter.uni-graz.at. Radiosonde profiles from the GRUAN network are accessible through page https://www.gruan.org/.

*Author contributions.* Petr Pisoft and Petr Sacha designed the study structure and working hypotheses. The algorithms were elaborated by Petr Pisoft who prepared the figures and draft of the manuscript, advised and supported in this work by Jiri Miksovsky, Peter Huszar, and Petr Sacha. Petr Sacha provided valuable feedback regarding the wave field stability and nonhydrostatic forcing. Barbara Scherllin-Pirscher prepared GPS RO profiles of the circulation field and together with Ulrich Foelsche provided discussion of the GPS RO retrieval and utilization. All authors contributed for submission and towards publication of the paper.

*Competing interests.* The authors declare that they have no conflicts of interest.

*Acknowledgements.* The presented work would not be possible without the utilized datasets. We are grateful to the UCAR/CDAAC, ROM-SAF, and WEGC RO processing team members. We would also like to thank the staff at the GRUAN sites at Lindenberg, Ny-Alesund, and Tateno for conducting the observations. This study was supported by GA CR under grant no. 16-01562J and project 7AMB16AT021 of the Czech Ministry of Education Youth and Sports. Collaboration of the Czech and Austrian team was supported by OeAD cooperation project CZ 06/2016. Petr Sacha was also supported by Spain government under the grant no. CGL2015-71575-P.





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





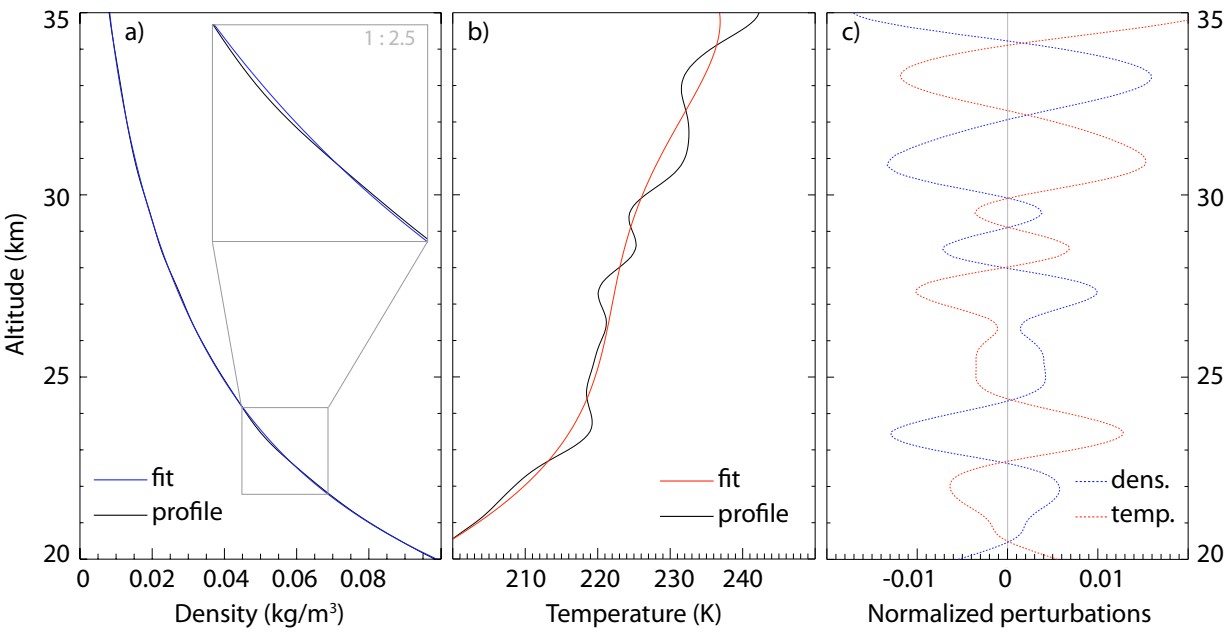

**Figure 1.** Example profile, background fit for density (a) and temperature (b), and normalized perturbations (c)





**Figure 2.** Power spectra density (PSD) of normalized perturbations and their differences for selected variables and datasets (indicated in the individual panels).

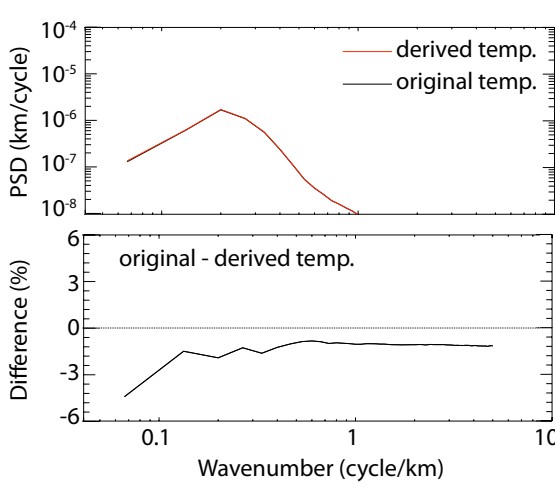

**Figure 3.** Power spectra density of normalized temperature perturbations and their differences for GPS RO and derived temperature profiles.






**Figure 4.** Power spectra density (PDS) of normalized temperature perturbations (a,c,e) and their differences (b,d,f) for radiosonde-measured and derived temperature profiles at the stations Lindenberg, Ny-Alesund and Tateno. Distributions of the PSD differences between the radiosondes and derived profiles are shown in panels g-i; the color-scale represents the number of profiles falling into a specific interval of perturbation differences and wave numbers.





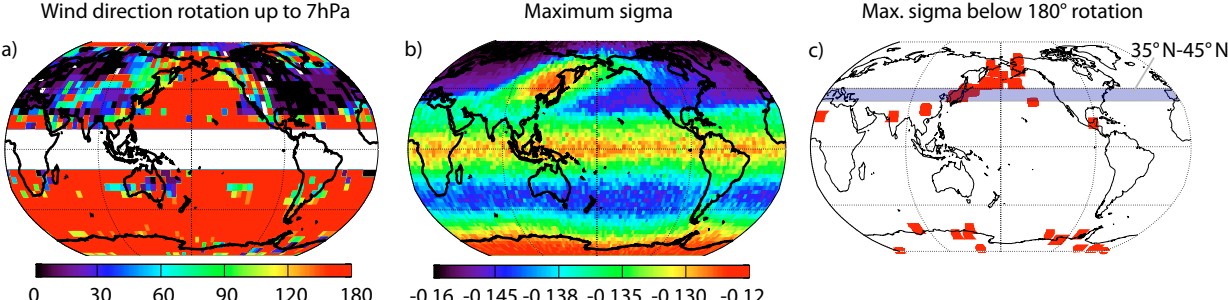

**Figure 5.** Vertical rotation of the wind direction between the surface and 7 hPa (a), distribution of maximum $\sigma^2$ values (b) and grid points where the maximum $\sigma^2$ values were detected below the critical rotation level (c). All the results are for the DJF season. The color scale represents the rotation in degrees (a) and $\sigma^2$ (b).

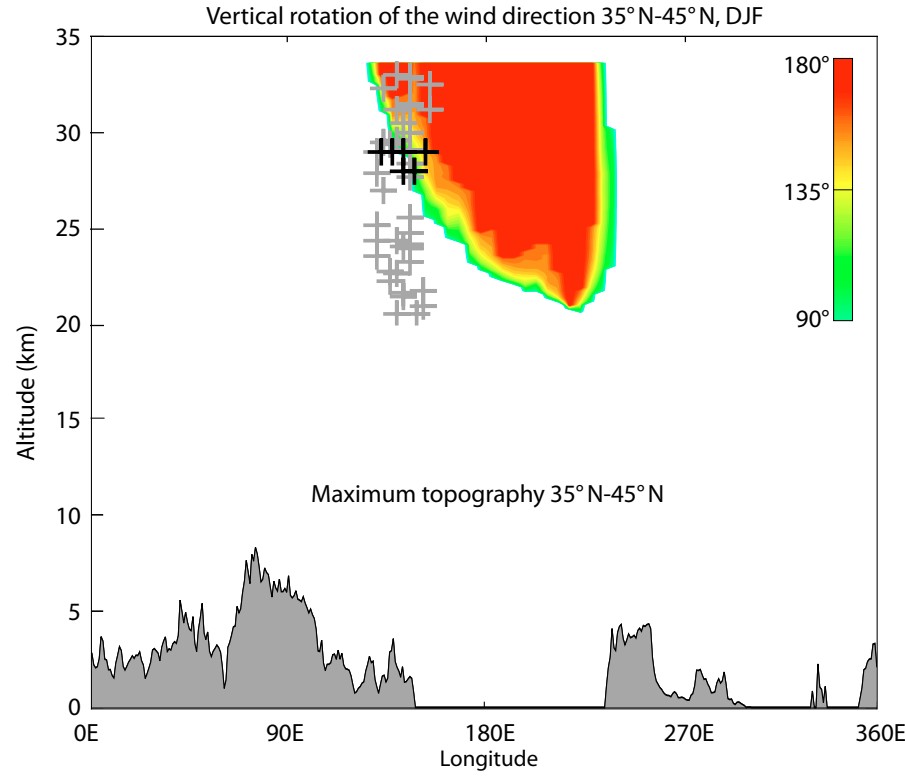

**Figure 6.** Vertical profile of the wind direction rotation between the surface and 7 hPa for the DJF season and 35° N–45° N latitudinal belt (top) and related maximum topography (bottom). The color-scale represents degrees of the rotation. The crosses represent the height of maximum $\sigma^2$ values (black represents the profile means, grey represents individual profiles).



**Figure 7.** Vertical profiles of the zonal wind speed (a), temperature (b), Richardson gradient number (c) and potential energy (d) for the DJF season and the 35° N–45° N latitudinal belt. The crosses represent the height of maximum $\sigma^2$ values (black represents profile means, grey represents individual profiles). The white line depicts the borders of the region where the vertical rotation of the wind direction exceeds 180°.