# Peer review of "Revisiting internal gravity waves analysis using GPS RO density profiles: comparison with temperature profiles and application for wave field stability study"

_Atmospheric Measurement Techniques, 2017_

## Referee Comment (RC1) · A. de la Torre (Referee) · 13 Oct 2017

Comments on:

**Revisiting internal gravity waves analysis using GPS RO density profiles: comparison with temperature profiles and application for wave field stability study**, by Petr Pisoft et al.

I appreciate the response given by the authors to my comments.

Regarding points 1) and 2) from the response to my previous comments, I understand that the authors consider that the hydrostatic hypothesis $\delta(p_o+p')/\delta z = -g(\rho_0+\rho')$ is equivalent to $\delta p'/\delta z = -g\rho'$, as far as $\delta p_o/\delta z = -g\rho_0$ may be safely applied to any background atmosphere. I have still two points that I would like to clarify: the spectra arising from GPS RO T or density data should then be expressed in the text as "apparent" or at least derived from "apparent" vertical or horizontal wavelengths measured from slanted soundings. The second point is a question: When you state in Šácha et al. (2014), after the hydrostatic hypothesis, that "...the whole group of nonhydrostatic IGWs is filtered out", I understand that one consequence of this is that in any GW climatology obtained from GPS RO T data, only hydrostatic and hydrostatic rotating aspect ratios could be detected. If this is correct, how could it be explained the systematic and clear hotspots from obvious orographic (non hydrostatic) origin above mountain regions, like at the southern tip of Southamerica, reported in a considerable number of papers in the last decade showing global distributions (and its variability) of GW energy?

---

## Referee Comment (RC2) · Anonymous Referee #1 · 21 Nov 2017

Reply to paper
Revisiting internal gravity waves analysis using GPS RO density profiles:
comparison with temperature profiles and application for wave field stability study

By P. Pišoft, P. Šácha, Jiri Miksovsky, Peter Huszar, Barbara Scherllin-Pirscher,
and Ulrich Foelsche

The manuscript is devoted to determination of characteristics of internal gravity waves (IGW) from the RO density vertical profiles. This interesting idea is being developed by the Authors in the course of several years.

1. The Authors have analyzed connection between IGW activity, wave breaking, background temperature, vertical rotation of zonal wind, its rotation with height) from GPS RO data.

2. The paper also presented a detailed study for the characterization of the regional wave field stability. The new selected stability parameters derived from the density profiles together with a study of the vertical rotation of the wind direction have been determined.

3. A potential IGW breaking area is detected in the western border of the Aleutian High where potential IGW breaking is detected.

4. The Authors' s results confirmed importance of the density profiles for IGW analysis. The Authors new message consists in using high-altitude air density in contrast to vertical temperature profiles for wave field stability study.

The content of the manuscript extends an applicable domain of the RO method and therefore the paper is important for AMT audience after minor revision.

Shortcomings.
For a reader it is difficult to understand the next incomprehensible statements:
1. "*The previously detected differences in the IGW spectra between dry temperature and density profiles are found only in the one specific data version*". I suppose that this should be removed from paper.
2. The Authors stated: "*The differences between temperature and density perturbations do not have any physical origin and there is no information loss of IGW activity due to the GPS RO retrieval*".Then the Authors claimed: "*We provide strong evidence that the differences in IGW perturbations between the real and retrieved temperature profiles (which are based on the assumption of hydrostatic balance) include a significant nonhydrostatic component that is present sporadically and might be either positive or negative…*"
These contradictions should be excluded (or explained carefully) in the manuscript.

---

## Author Response (AR1)

**Final response to the reviewers' comments of paper "Revisiting internal gravity waves analysis using GPS RO density profiles: comparison with temperature profiles and application for wave field stability study" by Petr Pisoft et al.**

We thank both reviewers for the positive judgment on our manuscript and their comments. We took all the reviewers' comments into account when preparing the revised version of the manuscript. The changes in the manuscript are highlighted in the version that is attached to this response.

Responses to particular referees comments are listed below.

Besides the changes connected to the referees comments we have also included a new information about affiliation of Petr Sacha that was missed in the original manuscript version.

**Reviewer #1**

1) *The spectra arising from GPS RO T or density data should then be expressed in the text as "apparent" or at least derived from "apparent" vertical or horizontal wavelengths measured from slanted soundings.*

Thank you for this comment, it is a very good point and we will incorporate this suggestion into the paper.

2) *When you state in Sacha et al.(2014), after the hydrostatic hypothesis, that "...the whole group of nonhydrostatic IGWs is filtered out", I understand that one consequence of this is that in any GW climatology obtained from GPS RO T data, only hydrostatic and hydrostatic rotating aspect ratios could be detected. If this is correct, how could it be explained the systematic and clear hotspots from obvious orographic (nonhydrostatic) origin above mountain regions, like at the southern tip of South America, reported in a considerable number of papers in the last decade showing global distributions (and its variability) of GW energy?*

In this paper, we intend to revise and correct this statement from Sacha et al., 2014. The hydrostatic temperature retrieval does not filter out any information. Instead, in case there are non-hydrostatic waves present, the density and temperature are not hydrostatically linked and the derived (GPS RO) temperature differs from the "real" temperature that would be observed directly. As pointed by the referee, this would be especially the case for the southern tip of South America.

Regarding the potential energy Ep for the southern tip of South America and other regions of a higher slope of IGW phase lines (higher Ep/kinetic energy of disturbances ratio), we refer to the discussion of Sacha et al. (2015), where it is noted that in such a region IGW activity can be overestimated using Ep from observations non-saturated spectra.

**Reviewer #2**

1) *"The previously detected differences in the IGW spectra between dry temperature and density profiles are found only in the one specific data version". I suppose that this should be removed from paper.*

Thank you for pointing out that the statement is not easily comprehensible. We will reformulate the sentence in the following way: "*we show that the differences in the IGW spectra between the dry temperature and density profiles that were described in the previous study as a general issue are in fact present in one specific data version only*".

2) *The Authors stated: "The differences between temperature and density perturbations do not have any physical origin and there is no information loss of IGW activity due to the GPS RO retrieval". Then the Authors claimed: "We provide strong evidence that the differences in IGW perturbations between the real and retrieved temperature profiles (which are based on the assumption of hydrostatic balance) include a significant nonhydrostatic component that is present sporadically and might be either positive or negative...". These contradictions should be excluded (or explained carefully) in the manuscript.*

Thank you for pointing out lack of clarity resulting in contradictions in our statements. The first statement is connected to the previous study where we speculated that the hydrostatic filtering is responsible for the differences in the IGW spectra between the dry temperature and density profiles. In the presented paper we show that those differences are in fact detected only in one specific data version, they do not have any physical origin and there is not the information loss of IGW activity that was suggested in the previous study. On the other hand in case of nonhydrostatic forcing, pressure and density are not hydrostatically linked and the induced dry density perturbations are connected with different temperature perturbations from those that are derived using the hydrostatic balance integration. Thus there is a difference between the real and retrieved temperature profiles (which are based on the assumption of hydrostatic balance) and the difference includes a significant nonhydrostatic component that is present sporadically and might be either positive or negative.

To avoid the potential contradictions in the abstract we will reformulate the first statement in the following way: *"
[revised manuscript text omitted]